# Using Smartphones for Indoor Fire Evacuation

**DOI:** 10.3390/ijerph19106061

**Published:** 2022-05-16

**Authors:** Tzu-Wen Kuo, Ching-Yuan Lin, Ying-Ji Chuang, Gary Li-Kai Hsiao

**Affiliations:** 1Department of Architecture, National Taiwan University of Science and Technology, Taipei 106335, Taiwan; linyuan@mail.ntust.edu.tw (C.-Y.L.); d9413005@gmail.com (Y.-J.C.); 2Department of Preparedness and Emergency Response, Office of Disaster Management, New Taipei City 220242, Taiwan

**Keywords:** smartphone, fire, evacuation, smartphone voice-guided evacuation system

## Abstract

Fire smoke reduces the visibility of emergency direction signs, rendering them ineffective for providing appropriate guidance along evacuation routes in a fire situation. This is problematic because civilians may select evacuation routes that expose them to smoke or fire in a burning building. This study proposed using a smartphone voice-guided evacuation system (SVGES) to provide alternative evacuation routes for civilians trapped at a fire scene. To verify the efficacy of the SVGES, experiments were conducted with 26 participants in a simulated fire scenario. The experimental results showed that when using the SVGES, the participants chose the safest evacuation route with a 100% successful evacuation rate.

## 1. Introduction

Innovative technologies have advanced considerably in the post-pandemic era. However, such advancements have not been widely applied to the field of fire evacuation. In the event of a fire incident, people are not always aware of where the ignition point is, which may lead to poor decisions locating and moving toward emergency exits during a fire evacuation [1]. The implementation of proper evacuation procedures can thus facilitate emergency management in buildings and environments where a fire incident occurs [2]. 

Currently, stationary active dynamic signage systems are the standard for indicating the direction in which people should move during an emergency [3,4,5,6,7,8,9]. Since these tend to be mounted on a wall or ceiling [10], smoke can easily affect visual guidance during a fire [11,12,13]. In fact, some research has shown that such systems are not an optimal solution for evacuation [14].

The ability to locate people quickly and accurately in buildings that are on fire is critical to the success of building fire emergency response operations [15]. Since indoor positioning systems are not yet widely applied worldwide [16], the orientation and route can only be identified by indoor signs. However, both visual and auditory information are important, especially for providing emergency information in many early warning systems [17].

Smartphone penetration has been forecasted to grow in the next few years. Smartphones can be considered an excellent platform that provides ubiquitous computing [18]. Therefore, the present study proposed a smartphone voice-guided evacuation system (SVGES) as a voice-based evacuation directional indicator that can be operated on a smartphone [19]. The SVGES provides route signage to guide people along the most appropriate evacuation route and away from the fire source [20], helping them to safely reach an emergency exit [21]. 

Research has recognized the practical value of smartphones as a tool for indoor pedestrian navigation [22,23], and the precise indoor positioning they can provide demonstrates their potential for future applications [24]. Particularly in dark environments, evacuations can be complicated by possible obstacles and nodes. Previous research showed that using a smartphone app for guidance in dark indoor environments can improve wayfinding efficiency [25].

The process for developing the SVGES proposed in the present study was based on the following three objectives: (1) design electronic route signs (ERSs) for indoor positioning and identifying the safest evacuation route; (2) design a smartphone app that uses voice, image, and text notifications to help evacuees choose the most appropriate evacuation route; and (3) verify the efficacy of the SVGES for aiding evacuation route choices through experiments.

## 2. Method

Intelligent systems are typically set up in centralized facilities and may be damaged during a fire. Thus, portable devices such as smartphones may be more reliable for distributed operation. For ease of use, we tested this with a commercially available Bluetooth 5.0 beacon. We adopted a guidance method used with movable road signs to solve the need to rely on precise positioning. Escape routes were predesigned by an architect in accordance with local regulations, and the strongest ERS Bluetooth beacon signal received by a smartphone was used as a positioning reference. The system was designed to handle changing environmental conditions as a fire develops, and to provide building occupants with turn-by-turn navigation guidance through their smartphone [26]. The ERSs were configured so that one or several fire points corresponded to a predesignated escape route. Each ERS on each route had its own auditory system, thus ensuring that low positioning accuracy would not cause any problems navigating the escape routes.

Experiments were conducted in a university classroom to determine whether the SVGES could guide evacuees in the correct direction. User acceptance and the ability of the app to reduce casualties were also examined. The equipment, participants, procedures, and test scenarios are described in the following sections. 

### 2.1. SVGES

The SVGES included a preset software program that generates the safest evacuation route according to different building floor plans. These evacuation routes were predesigned by the architect in accordance with local regulations. The SVGES provides dynamic recommendations for avoiding fire and smoke, based on the safest, shortest distance. Development and operation of the SVGES involved the following hardware and software components:Smartphone;Smoke detectors;ERSs
(a)Battery;(b)Two beacons;(c)Photosensitive components;Smartphone app.

The SVGES can communicate with a smartphone in the situation of a fire, providing guidance along evacuation routes via voice, image, and text notifications. Figure 1 shows a flow chart of the SVGES.

#### 2.1.1. ERS

Each ERS comprised a photoresistor and two beacons mounted outside of a smoke detector, with each smoke detector assigned a specific detector beacon number to indicate its spatial location. These evacuation routes were pre–designed by an architect in accordance with local regulations. In the event of a fire, the smoke detector nearest to the fire activates first. When the photosensitive component on the ERS is triggered by the LED on the smoke detector, the beacon starts transmitting a signal via Bluetooth. Based on these signals, the app can detect the fire source and indicate the safest evacuation route. Figure 2 shows a diagram of an ERS fitted to a smoke detector.

#### 2.1.2. Smartphone App

The SVGES app is designed to operate on Android smartphones. It was programmed to identify a fire source and determine the safest evacuation route through the following steps: (1) track the ERS Bluetooth signals every second; (2) determine the location of the user’s smartphone relative to the ERSs; (3) identify the location of the fire source relative to the user; (4) determine the safest evacuation route; and (5) notify evacuees which direction they should move via voice, images, and text.

In the event of a fire being detected, a designated shortest evacuation route is calculated for each ERS, and each ERS transmits its own voice, image, and text notifications to guide evacuees along the safest evacuation route via the app. Figure 3 depicts the app user interface showing the safest evacuation route.

### 2.2. Participants of the Experiment

A total of 26 participants (14 men, 12 women) took part in the experiment. Their age distribution was 20–50 years, with an average of 32 years. All of them were unfamiliar with the experimental environment, and this was the first time they had been in the experimental space. Regardless of the number of trials, the participants remained unaware of the location of the fire source. 

### 2.3. Experimental Environment

The experimental environment was on the second-floor basement of a university building. Figure 4 shows the floor plan. All rooms on the floor had a single door as the exit. The floor had one corridor with two exits. Each room and the corridor were regarded as nodes for choosing a direction for a specific evacuation route. 

### 2.4. Experimental Procedure

Before the experiment, the participants provided informed consent and were informed that they would be taking part in a simulated fire evacuation route experiment, the aim of which was to study how to choose the best evacuation route away from the fire source. Participants were asked to act as if they had actually encountered a fire and to leave the room in which they started as quickly as possible in order to find an exit to evacuate.

The researchers designated a fire source along one of the evacuation routes. Two experiments were performed. In the first experiment, the participants were required to evacuate without using their smartphones and to freely select an evacuation route. In both experiments, the evacuation progress of each participant was recorded until the end of the experiment, including their chosen route and the time elapsed from when they left the room they started in. 

Prior to the second experiment, the participants were informed that the fire source might be in a different location, and that they needed to use the SVGES app to assist them with their evacuation route decisions. The designated location of the fire source in this experiment was actually identical to that in the first experiment, but the participants were unaware of this. 

At the beginning of the second experiment, the fire alarm system and SVGES were activated, and the participants’ smartphones scanned the ERS located in the starting room. The smartphone app then provided warnings of where the fire was located and notified the participants of the safest evacuation route via voice messages. The user interface displayed text messages such as “move forward”, and this was supplemented with images.

Following the evacuation route directions given by the app, the participants began by moving to the door of their starting room. The app then scanned the other ERSs located on the ceiling of the corridors to detect the location of the fire source and then provided appropriate guidance to the participants. For example, if the fire source was located to the left of the starting room, the app instructed them to “go right”. If participants did not follow the notifications from the app (e.g., they went left when they were instructed to go right), the app would detect that they were moving away from the safest evacuation route. If they moved further away from the recommended route, the app would instruct them to “go back” via voice, image, and text notifications.

Referring to Figure 4, the app was able to detect which direction the participants were moving in based on their position relative to the ERSs. The signal from ERS 1 (S1 in the figure) was used to identify their starting point. If a participant turned left as they exited the starting room, their position would be closer to ERS 3 (S3) and further from ERS 2 (S2), indicating to the SVGES that they were actually moving toward the fire source. Thus, the SVGES could determine that the participant was moving away from the safest evacuation route and then inform them to turn around and move in the other direction.

## 3. Results and Discussion

This research proposed an SVGES that provides the safest evacuation route in a fire building by using voice, image, and text messages. In the experiment, the SVGES was implemented in an experimental environment to verify the evacuation route choices made by the participants in a fire situation.

In the first experiment, the participants freely selected their evacuation route without using the SVGES (no SVGES). The second experiment was performed with the participants using the SVGES and following the notifications from the SVGES when choosing their evacuation route. If a participant’s evacuation route was toward the exit near the designated fire source, this was defined as a failed evacuation. If the participant chose an evacuation route that was toward the exit that was away from the designated fire source, this was defined as a successful evacuation. In the first experiment, the successful evacuation rate was 58%. When the SVGES was used, the successful evacuation rate was 100%. Previous studies have found that the presence of more nodes during an evacuation can lead to higher failed evacuation rates [27,28]. Using technology to guide wayfinding behavior in a fire scene is critical because blind wayfinding under thick smoke is highly dangerous. In the process of the evacuation experiment, Figure 4 shows that although the floor plan was very basic, there were still multiple nodes for evacuees to decide on a possible direction to move in to reach safety. In a more complex environment with additional nodes, the probability of a failed evacuation would be substantially higher. 

Previous studies have not focused on using smartphones for controlling evacuation signs based on the positioning of devices, and they have overlooked the importance of providing evacuee-specific voice guidance [29]. An efficient evacuation system could reduce the stress levels of evacuees and significantly increase the survival rate during a fire evacuation [30]. In our experiments, we found that the SVGES improved evacuation efficiency. In an actual building fire, the SVGES might reduce the number of casualties.

Although it may be difficult to view the appropriate evacuation direction indicated on the smartphone when thick smoke is present, the app’s voice notifications overcome this by also announcing the direction in which evacuees should move in to reach safety. Thus, the SVGES may help evacuees even in dark spaces with thick smoke.

Path selection in the event of a fire is key to success or failure. Recent research on guided evacuation has typically focused only on visual guidance while ignoring the importance of auditory guidance. Our experimental results clearly indicate that using the proposed SVGES can help evacuees choose the appropriate direction to move toward in order to reach safety. 

Leader–follower behavior can adversely affect evacuation route decisions during a fire evacuation [31,32,33]. The proposed SVGES may overcome this problem by providing clear directions for evacuees, ensuring that they move away from a fire source in what might be an unfamiliar location with many nodes.

The experiments in this study were conducted using a single-path node. On the basis of probability, fire situations involving many intersection nodes along an evacuation path are likely to result in a higher rate of evacuation failure when the SVGES is not used. 

Previous studies have measured the required safe egress time against available safe egress time (Equation (1)) when investigating the success probability of escape and evacuation, and they have found that the correct selection of a node direction along a path is critical to success [34,35]. Therefore, the present study recommends that whether the selected node direction along a path is correct or not should be considered in this calculation, as shown in Equation (2).
REST = Td + Ta + Tr + Tt ≤ ASET(1)

This study found that the mathematical formula after adding parameters should be recommended (Equation (2)):REST = Td + Ta + Tr + Tt/J^n^ ≤ ASET(2)

REST: Required safety egress time;

T_d_: Fire detection time;

T_a_: Fire alarm time (warning time);

T_r_: Recognition time and response time;

T_t_: Evacuation time (travel time);

n: Number of turn nodes on the escape path (n > 0);

J: Correct path node direction (if correct = 1; otherwise, 0);

ASET: Available safe egress time.

## 4. Conclusions

Fire equipment installed in a building might not ensure absolute safety during an evacuation. The present study proposed an SVGES and designed ERSs for indoor positioning, enabling the SVGES to direct evacuees along the safest evacuation route via voice, image, and text notifications on a smartphone app. Experiments were conducted to verify the efficacy of using the SVGES in a fire building. When 26 participants evacuated using the SVGES, the successful evacuation rate was 100%. When the SVGES was not used, the successful evacuation rate was only 58%.

The results demonstrate the success of using the SVGES in a single-room environment and indicate that it is feasible for use in a fire evacuation. Theoretically, it can guide any number of people to safety in a fire event; however, if more nodes are present, this may adversely affect evacuation route decisions, potentially resulting in a higher rate of failed evacuations. This emphasizes the importance of making the correct decision at nodes along an evacuation path. Future studies should investigate the proposed SVGES being implemented in environments that are more complex and contain multiple fire sources to identify and overcome deficiencies in the current research.

## Figures and Tables

**Figure 1 ijerph-19-06061-f001:**
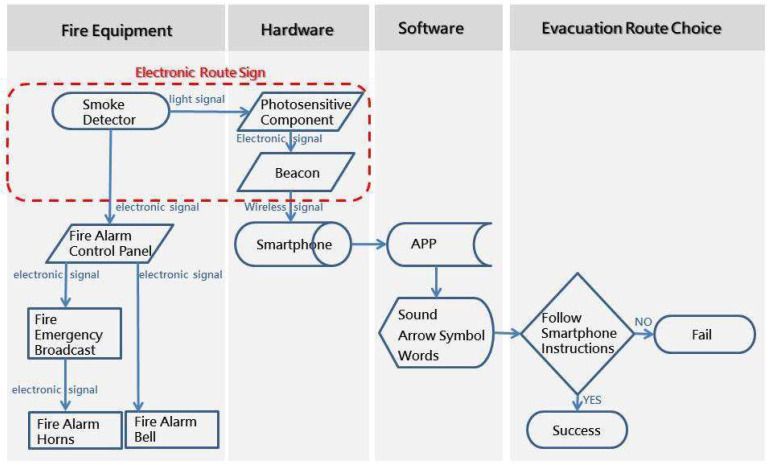
Flow chart of SVGES.

**Figure 2 ijerph-19-06061-f002:**
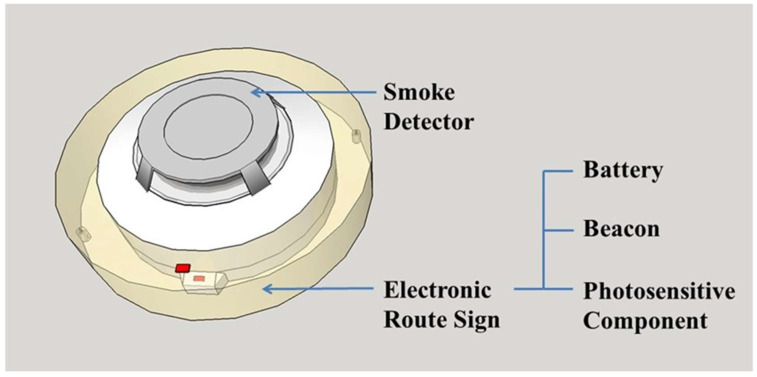
Diagram of a smoke detector fitted with an ERS.

**Figure 3 ijerph-19-06061-f003:**
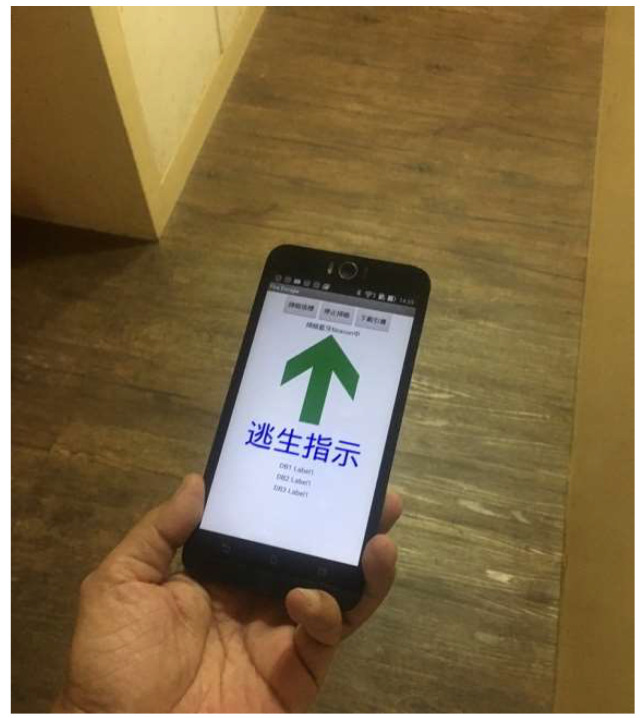
Smartphone app user interface showing the safest evacuation route. (Chinese in this figure means Escape instructions).

**Figure 4 ijerph-19-06061-f004:**
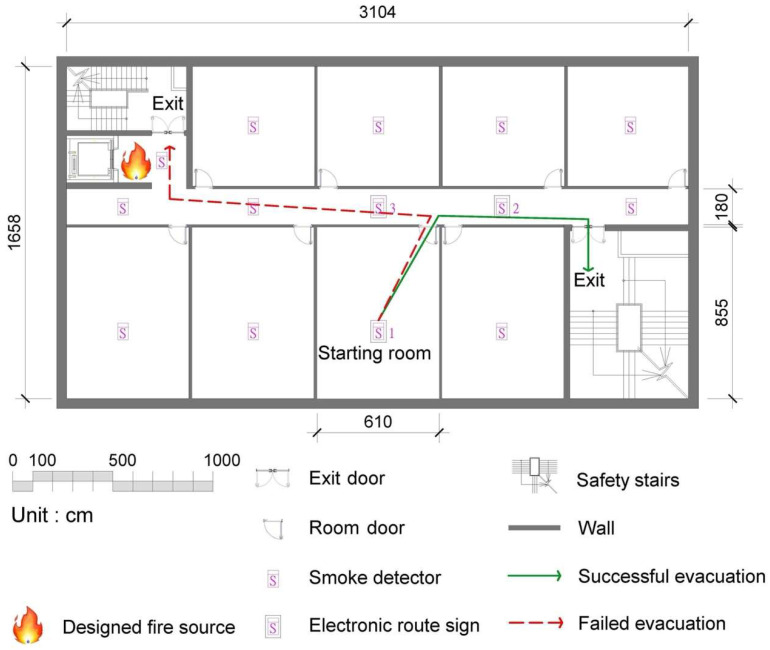
Floor plan of the experimental environment.

## Data Availability

The datasets generated and analyzed during the current study are not publicly available due to the restriction of the library surveyed in the study, but are available from the corresponding author on reasonable request.

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
