# Peer review of "Using Smartphones for Indoor Fire Evacuation"

_ijerph, 2022, doi:10.3390/ijerph19106061_

Round 1

Reviewer 1 Report

Strengths:
 (+) The problem is important.
 (+) The figures are appropriate.
 (+)
 (+)

Weaknesses:
 (-) There are English issues.
 (-) References are inadequate.
 (-) The introduction must be improved.
 (-) The related work section must be enhanced.
 (-) The method is too simple.
 (-) Experimental evaluation must be improved.
 (-) There are issues related to the problem definition.
 (-) Some improvements are needed in the description of the method.
 (-)
 (-)

==== ENGLISH ==== 

The paper has several typos. Authors need to proofread the paper to eliminate all of them.

Some sentences are too long. Generally, it is better to write short sentences with one idea per sentence.

==== REFERENCES ==== 

The literature review is incomplete. Several relevant references are missing.

==== INTRODUCTION ==== 

The authors should add more references in the introduction to support the claims. Such as: Features of mobile apps for people with autism in a post covid-19 scenario: current status and recommendations for apps using AI, to investigate and enhance the literature. 

The authors need to better explain the context of this research, including why the research problem is important.

Contributions should be highlighted more. It should be made clear what is novel and how it addresses the limitations of prior work. 

==== RELATED WORK ==== 

The related work section is not well organized. Authors must try to categorize the papers and present them in a logical way.

The authors should add a table that compares the key characteristics of prior work to highlight their differences and limitations. The authors may also consider adding a line in the table to describe the proposed solution.

==== PROBLEM DEFINITION ==== 

The authors should add a clear and detailed problem definition. 

The authors should add an example to illustrate the problem definition. 

==== METHOD ==== 

A novel solution is presented but it is important to better explain the design decisions (e.g. why the solution is designed like that)

It is necessary to discuss the complexity of the proposed solution.

==== EXPERIMENTS ==== 

The experiments should be updated to include some comparisons with newer studies. 

Author Response

Dear Reviewer,

We really appreciate your valuable comments and suggestions.

Please check the attachment for responses of the comments.

Reviewer 2 Report

The Authors propose a solution for guiding people along indoor evacuation routes in case of fire. The key contribution is based on the use of dedicated electronic route signs and a mobile app. The solution is described in a very general manner, without any details on the key problems related to the guidance of people in unknown environment. The Authors fail to provide any information on the positioning and route discovery algorithms/methods they use. Also, the successful implementation of the solution assumes the installation of dedicated software by the users, and installation of new class of electronic route signs. Although the Authors present some results of system verification (test with participation of a group of 26 users), the paper provides no discussion on factors that may influence the performance of the proposed solution.

Section 2 Methods

The section contains both the description of the proposed solution as well as the description of the setup used during the experimental verification of the solution. I would recommend to include in this section more detail on the proposed solution, and the subsections on the experiments move to Section 3 Results. This section (Section 2) fail to provide any details on the positioning and routing methods used in the systems – these are vital components of the entire solution, and its overall performance strongly depends on the algorithms used. Are these algorithms based on well-known solutions or have been developed by the Authors? Can these algorithms be applied to any environment?

The mobile app – what is the applicability of the app in limited (or no) visibility conditions?

No details on the design of the ERS are provided. What was the motivation for the selection of Bluetooth communication protocol (and which version)? The Authors show that the ERS may be battery operated – how the use of frequent Bluetooth transmission affects the battery life?

Section 3

It is not clear how the decisions on selection of the evacuation route were made during the first phase of the experiment.

Were the participants of the second phase the same people as during the first phase (without SVGES)?

Sections 3/4

The Authors have not compared the obtained results against any competing solution, and rely only on the results of their own experiments. It recommended to compare the system functionality and its technical parameters against other state of the art solutions. It is also recommended to extend the experimental part to include more details on more technical system performance evaluation, not only showing that in works in relatively simple scenario.

Author Response

(The authors gave the same response as above.)

Reviewer 3 Report

The paper presents a very interesting idea of using smartphones to support indoor fire evacuation. It is really great issue, because many casualties are caused by improper choice of escape route due to of getting lost in a building, worsened visibility, stress or general lack of the understanding the situation (for instance the fire at Dusseldorf airport in 1996). However, the work can be improved. The list of detailed comments is as follows:

  1. Figures 2 and 3 can be merged – such level of detail is unnecessary. But some information on algorithm, which determines the best evacuation route would be valuable.
  2. How does the system operate? Are the plans of the building stored locally with the app, or are the tips on the evacuation transmitted from the stationary part of the system? In the second case the system would be much universal. This is a remark of a minor importance, however it may be crucial for the common system application.
  3. Were both experiments carried out with the same spatial arrangement of rooms, fire position and evacuees’ starting point? Was it possible, the evacuee had learnt the best way to escape and that fact could impact on the results?
  4. It would be better to carry out a number of experiments with different arrangements and then compare the results. However, I’m aware it could be impossible at this stage of the work.
  5. Is your system able to dynamically change the recommended evacuation route in a case of violent fire development?
  6. I wonder about the legal consequences of such an event: the escape is almost impossible, but your system has found the best way of evacuation, hence it poses a very high likelihood of health danger or even life lose. So, you do your best, but somebody may accuse the system for causing the danger. This is similar to the responsibility in a case of the autonomous vehicles

Author Response

(The authors gave the same response as above.)

Round 2

Reviewer 1 Report

Addressed comments. 

Author Response

Dear Reviewer,

We really appreciate your valuable comments and suggestions.

The manuscript has been revised.

Reviewer 2 Report

The Authors have addressed majority of issues raised in the first round of reviews. Thank you. However, the paper still fails to answer what are the positioning methods implemented in the system. The idea of using Bluetooth beacons (or radio beacons in general) for positioning is not new and numerous papers can be found on various approaches to indoor positioning as well as describing a wide range of problems related to position estimation in such systems. The problem of positioning beacomes vital especially in low visibility conditions (what is not unusual in case of fire).

Thus, I recommend including more details on:

  • how the position of the mobile is determined? Is it based only on the information that the mobile is within the range of the ERS transmitter? And what in case the ERSs have overalpping coverage areas (what is typical in radio systems)? How these problems are solved in the system?
  • what are the assumptions for deployment of ERSs in the building (e.g. density)?

Author Response

Dear Reviewer,

We really appreciate your valuable comments and suggestions.

The manuscript has been revised.

Please check the attachment for responses of the comments.

Reviewer 3 Report

Since the Authors have addressed all my remarks I have no further comments

Author Response

(The authors gave the same response as above.)
